# Regulatory Mechanisms of Epigenetic miRNA Relationships in Human Cancer and Potential as Therapeutic Targets

**DOI:** 10.3390/cancers12102922

**Published:** 2020-10-11

**Authors:** K. M. Taufiqul Arif, Esther K. Elliott, Larisa M. Haupt, Lyn R. Griffiths

**Affiliations:** Centre for Genomics and Personalised Health, Genomics Research Centre, School of Biomedical Sciences, Institute of Health and Biomedical Innovation, Queensland University of Technology (QUT), 60 Musk Ave., Kelvin Grove, Queensland 4059, Australia; kmtaufiqul.arif@hdr.qut.edu.au (K.M.T.A.); esther.elliott@hdr.qut.edu.au (E.K.E.); larisa.haupt@qut.edu.au (L.M.H.)

**Keywords:** epigenetics, microRNA, cancer, bioinformatics, therapeutics

## Abstract

**Simple Summary:**

By the virtue of targeting multiple genes, a microRNA (miRNA) can infer variable consequences on tumorigenesis by appearing as both a tumour suppressor and oncogene. miRNAs can regulate gene expression by modulating genome-wide epigenetic status of genes that are involved in various cancers. These miRNAs perform direct inhibition of key mediators of the epigenetic machinery, such as DNA methyltransferases (DNMTs) and histone deacetylases (HDACs) genes. Along with miRNAs gene expression, similar to other protein-coding genes, miRNAs are also controlled by epigenetic mechanisms. Overall, this reciprocal interaction between the miRNAs and the epigenetic architecture is significantly implicated in the aberrant expression of miRNAs detected in various human cancers. Comprehensive knowledge of the miRNA-epigenetic dynamics in cancer is essential for the discovery of novel anticancer therapeutics.

**Abstract:**

Initiation and progression of cancer are under both genetic and epigenetic regulation. Epigenetic modifications including alterations in DNA methylation, RNA and histone modifications can lead to microRNA (miRNA) gene dysregulation and malignant cellular transformation and are hereditary and reversible. miRNAs are small non-coding RNAs which regulate the expression of specific target genes through degradation or inhibition of translation of the target mRNA. miRNAs can target epigenetic modifier enzymes involved in epigenetic modulation, establishing a trilateral regulatory “epi–miR–epi” feedback circuit. The intricate association between miRNAs and the epigenetic architecture is an important feature through which to monitor gene expression profiles in cancer. This review summarises the involvement of epigenetically regulated miRNAs and miRNA-mediated epigenetic modulations in various cancers. In addition, the application of bioinformatics tools to study these networks and the use of therapeutic miRNAs for the treatment of cancer are also reviewed. A comprehensive interpretation of these mechanisms and the interwoven bond between miRNAs and epigenetics is crucial for understanding how the human epigenome is maintained, how aberrant miRNA expression can contribute to tumorigenesis and how knowledge of these factors can be translated into diagnostic and therapeutic tool development.

## 1. Introduction

microRNAs (miRNAs) consist of short sequences of non-coding RNA which regulate translation and expression of specific target genes. Currently, there are over 2500 miRNAs referenced on the global micro RNA database, miRbase [1]. The role of miRNAs is to negatively regulate gene expression through binding to the target mRNA to cause transcriptional repression and/or mRNA degradation without modifying the gene sequence. This occurs through a miRNA recognising their target mRNA using either 2–7 nucleotides (partially complementary) or 6–8 nucleotides at the 5′ end of the miRNA annotated as the “seed region”. The exact manner in which protein translation is downregulated by miRNAs is still not clear and may be due to either mRNA degradation and translation inhibition or a combination of these events [2]. miRNA expression is regulated during hematopoietic cell development and differentiation, with these miRNAs playing a direct regulatory role in processes such as cellular proliferation, differentiation, migration and apoptosis [3,4]. 

Biogenesis and expression of miRNAs are known to be regulated by epigenetic modifications such as DNA methylation, RNA alterations, and histone modifications, with dysregulation of miRNAs being a hallmark of cancer initiation and metastasis [5]. It has also been established that miRNAs control the expression of epigenetic regulators, including DNA methyltransferases and histone deacetylases [6]. miRNAs are involved in complex double-negative feedback loops where miRNA inhibition of an epigenetic regulator is then controlled at the epigenetic level by the same regulator. This miRNA–epigenetic feedback loop has a significant influence on gene expression levels, and dysregulation of the feedback loop can disrupt normal physiological processes, resulting in disease [7]. It is clear that miRNA–gene associations are not linear, hence, functional heterogeneity of a single miRNA across cell types, tissues and disease stages, increases the degree of difficulty in discerning the direct functional pathways regulated by any miRNA [8]. Aberrant miRNA profiling showing altered regulating factors such as cellular proliferation, and migration has been described in many cancers with the majority showing decreased miRNA expression levels in tumour cells in comparison to normal tissue [9]. 

The first miRNAs to be correlated with cancer were miR-15 and miR-16 in B-cell leukaemia [10]. Since then, the biogenesis of miRNA and target genes such as tumour suppressor and oncogenes has been well established [3,4,9,11,12] with the aid of both experimental and computational analyses; however, further validation using experimental analyses is required if we are to accurately understand the role of miRNA in the multiple functional pathways mediating cancers [3,5,7,9,13,14,15]. Identifying master regulatory miRNAs which regulate both the target mRNA as well as other miRNAs in either the same or a different pathway could provide early prognostic disease biomarkers for both solid tumours and haematological malignancies [9,16,17]. The degree of protein repression by one miRNA can range from mild (2-fold) to significant repression, highlighting the need for better experimental validation. RNase III Drosha is the core RNA-specific nuclease that executes the initiation step of miRNA processing in the nucleus. The resulting miRNAs then regulate the degree of gene repression by the association of enhanced processing of ribosomal RNA precursors by a nuclear dsRNA ribonuclease called Drosha to produce efficient RNA-induced silencing complex (RISC)-induced cleavage of complementary mRNA, resulting in enhanced mRNA repression [18]. Recent studies have shown that a large number of mature miRNAs are methylated at cytosines and adenines in cancer cell lines, as well as human tissues and serum, potentially providing a more accurate cancer diagnostic tool than miRNA expression [19].

The epigenome (the collection of chemical compounds that interact with and regulate the genome) consists of chemical compounds and proteins which regulate gene expression and protein production within cells. It can also be subjected to miRNA-mediated and other posttranscriptional alterations which result in dysregulated miRNA signatures relating to enhanced oncogene expression and downregulation of tumour suppressor genes, leading to tumorigenesis and cancer progression [20]. The epigenome is normally reversible; therefore, miRNA dysregulation might be predictive of cancer development when miRNA alterations become irreversible [21]. Epigenetic markers can also relate to clinical prognosis and may provide a tool for early detection and cancer treatment. By directly targeting miRNAs or epigenetic machinery, malignancy could potentially be treated using therapeutic agents [11,21,22,23] such as those discussed in this review. A considerable number of studies have demonstrated the orchestrated role of epigenetics and miRNA in diverse cellular processes and complex diseases like cancer. Unfortunately, such evidence is dispersed in the huge body of literature, making it difficult for researchers to investigate their reciprocal regulations. One way to achieve this is thorough computational data processing applications of the experimentally proven interactions between epigenetic modifications and miRNAs that can be stored as a searchable database [24,25]. Such information can provide invaluable information to better understand the molecular mechanisms of “epi–miR–epi” in cancer and encourage targeted research toward epigenetics- or miRNA-related drug development. 

## 2. Epigenetic Regulators of Cancer

### 2.1. DNA Methylation and miRNA Regulation

Aberrant expression of miRNAs significantly affects the gene regulatory mechanisms implicated in cancer development with miRNAs having the intricate ability to act as both oncogenes (oncomiRs) and suppressors [26]. It is an important fact to consider that one miRNA can regulate numerous genes, and can be targeted by several miRNAs [27]. Increasing evidence has indicated that miRNAs expression is under the control of epigenetic regulation, similar to DNA methylation (hyper and hypo), RNA modification, and post-translational modification of histones [28,29,30,31,32,33]. Interestingly, miRNAs contribute to epigenetic modulation by targeting epigenetic modifiers [34,35]. We have summarised the epigenetic regulation of miRNAs in haem-malignancies and solid tumours (Appendix A), as well as the miRNAs that affect epigenetic regulation and have been reported in various cancers (Table 1).

Disrupted DNA methylation in miRNA loci often leads to downregulation of the miRNA and a greater likelihood of displaying a malignant phenotype. This is seen when miRNAs are more highly expressed and have greater sequence conservation when flanking regions surrounding the miRNA coding sequence are highly methylated [4,9,11]. miRNA target gene promoters are often negatively correlated with DNA hypomethylation, and DNA methylation is regulated by three catalytically active DNA methyltransferases (DNMTs): DNMT1, DNMT3a, and DNMT3b [71]. The majority of studies on DNMT3a/b and miRNAs have primarily focused to date on the regulation of DNMT3a/b targets. However, more recently direct miRNA targeting of DNMT3a/b has shown the potential for both oncogenic [72] and tumour suppressor activities in the progression of cancer such as breast cancer [73].

A well-known miRNA which is often epigenetically regulated is miR-9. Its expression has been shown to be associated with hypermethylation of a cytosine and guanine separated by a phosphate (CpG) island in the miR-9 loci [74,75,76]. miR-9 hypermethylation is seen in many cancers including solid tumours such as breast, colon, and pancreas along with haematological malignancies like acute lymphoblastic leukaemia [75,77,78,79]. It has been suggested that epigenetic silencing of the miR-9 loci as a result of hypermethylation is often an early disease-associated event in breast carcinogenesis [75]. Other commonly hypermethylated miRNAs include miR-92 and miR-29b which target the TET gene family and act as oncogenic miRNAs (oncomiRs) causing reactivation of silenced oncogenes [80,81,82,83]. 

A number of miRNAs aberrantly silenced are involved in DNA methylation, histone acetylation, H3K4me3 and H3ac modifications and have been identified in haem-malignancies (Appendix A) and bladder, cervical, colorectal, gastric, hepatocellular, lung, melanoma, pancreatic and prostate cancers (Appendix A). Some of these miRNAs, including let-7a, miR-9, miR-34b-c, miR-124a, miR-127, miR-129, miR-137, miR-148a, miR-152, miR-203, miR-205, the miR-200 family and miR-375, have been frequently reported in several cancers (Appendix A) [40,42,46,47,68,69,84,85,86,87,88,89,90,91,92,93,94,95,96,97,98,99]. Among these, miR-9, miR-34b-c and miR-148a are frequently hypermethylated in aggressive tumours, with this feature proposed as a possible DNA methylation signature for metastasis [66,100,101,102,103]. In breast cancer, epigenetic regulators like DNMTs (1/3), HDAC, JARID1B (histone H3 lysine 4 demethylase) and sp1 (*Sp1 Transcription Factor*) cause both higher and lower expression of several miRNAs including miR-124.3, miR148a, miR-375, miR-152, members of miR-200, let-7 and miR-34 families [5,36,85,87,104,105,106,107,108,109]. This may be important clinically, with lower expression levels of miR-29c, miR-148a, miR-148b, miR-26a, miR-26b, and miR-203 demonstrated to contribute to DNMT3b overexpression in hypermethylated breast cancer cell lines (Hs578T, HCC1937 and SUM185). In contrast, knockdown of miR-148b, miR-26b, or miR-29c in non-hypermethylated breast cancer cell lines (MDA-MB-468, MDA-MB-415, and BT20) showed increased DNMT3b mRNA levels [40]. 

These “oncomiRs” may be switched on by abnormal CpG hypomethylation, an indicator of oncogene activation [110]. As discussed earlier, several studies using a range of cell lines have indicated overexpression of numerous miRNAs following treatment with epigenetic drugs [111,112,113]. The let-7a-3 locus is hypermethylated in normal human lung tissue, but in some lung adenocarcinomas, it becomes hypomethylated and overexpressed [114]. miRNA-mediated deregulation of DNMTs has been observed in several cancers. In lung cancer, the miR-29 family that targets DNMT3a-b was shown to be downregulated [30]. In addition, in hepatocellular carcinoma, miR-29a regulates both the DNA DNMT1 and DNMT3b [40]. miR-148 has also been identified to regulate a DNMT3b splice variant through binding to the coding region in gastric cancers [47]. miR-449a was found to be involved in cell growth and viability regulation through repressing the expression of HDAC-1 in prostate cancer cells [67]. Another miRNA frequently reported in a number of cancers is miR-101 [48,49,68]. The histone methyltransferase EZH2 has been shown to be a direct post-transcriptional target of miR-101, with miR-101 downregulation by VEGF resulting in overexpression of EZH2 in angiogenic endothelial cells [49]. 

### 2.2. Histone Acetylation and Deacetylation

The chromatin structure is comprised of DNA and histones made up of a chain of 147 nucleotides wrapped around a histone octamer consisting of two copies of each of four histones: H2A, H2B, H3, and H4. Chromatin features are involved in both activation and repression of transcription [115,116,117]. A relationship between miRNA biogenesis and chromatin features around pre-miRNA genomic regions has also been previously reported in which genes located within active chromatin regions have a higher probability of being targeted by miRNAs [118]. The promoters of miRNA target genes have also been shown to be preferentially located in chromatin domains [119]. Additional to the influence of chromatin on miRNA target gene regulation are the impacts of posttranslational modifications of histones through events such as phosphorylation of serine or threonine residues, acetylation and deacetylation of lysine and methylation of lysine or arginine. Histone modifications that regulate epigenetics include methylation (HDM) and acetylation (HDA) which affect DNA accessibility typically modulated by histone acetyltransferase (HAT) and histone deacetylase (HDAC). HDAC causes condensation of the chromatin structure, which prevents binding of transcription factors or proteins to the DNA strand, resulting in gene silencing [120,121]. HDAC inhibits differentiation pathways critical to cellular development and differentiation, as well as the maturation of antibody response [122]. In cancer, demethylation of genes by HDMs often results in upregulation of genes involved in cellular proliferation, migration, and invasion. This provides an opportunity for miRNA-based agents which could regulate proteins in these pathways, thereby inhibiting malignant cellular proliferation [123].

## 3. Epigenetic Regulation by miRNAs in Cancer

miRNAs can also directly alter epigenetic regulation by post-transcriptionally suppressing the mRNAs of genes involved in the deposition of epigenetic marks (Table 1). Fabbri et al. demonstrated that the miR-29 family targets DNMT3a and DNMT3b, which indirectly controls genome-wide de novo DNA methylation [62]. Soon after, it was validated that miR-29b induces DNA hypomethylation universally in acute myeloid leukaemia (AML) by direct downregulation of DNMT3a and DNMT3b and indirect repression of DNMT1 [37]. DNMT3a is also a direct target of miR-143 [43]. The epigenetic-cally modulated miR-148a suppresses particular isoforms of DNMT3b by targeting the coding sequence [41,47]. Furthermore, DNMT1 has been confirmed as a target of miR-148a and miR-152 [41]. Such a direct effect on the DNA methylation machinery by epigenetically regulated miRNAs could be an indication of an important feedback regulatory mechanism. Several reports have demonstrated that DNMT3a and DNMT3b are fre-quently overexpressed in cancers with poor prognosis, and targets of the miR-29 family, miR-101, miR-143, mir-148a and miR-152 have been reported in various forms of cancer [37,41,43,45,46,47,51,59,61,62,65,66]. Therefore, it appears that all DNMTs are a direct or indirect target of a subset of miRNAs and perform specific functions in the refinement of the expression levels of these crucial epigenetic regulators. As such, dysregulated miRNA expression may contribute to the widespread and often inconsistent changes in DNA methylation patterns detected in cancers, causing both hypo- and hyper-methylation of certain genes and/or regions of the genome [124].

miRNAs can also directly alter epigenetic regulation by post-transcriptionally suppressing the mRNAs of genes involved in the deposition of epigenetic marks (Table 1). Fabbri et al. demonstrated that the miR-29 family targets DNMT3a and DNMT3b, which indirectly controls genome-wide de novo DNA methylation [62]. Soon after, it was validated that miR-29b induces DNA hypomethylation universally in acute myeloid leukaemia (AML) by direct downregulation of DNMT3a and DNMT3b and indirect repression of DNMT1 [37]. DNMT3a is also a direct target of miR-143 [43]. The epigenetic-cally modulated miR-148a suppresses particular isoforms of DNMT3b by targeting the coding sequence [41,47]. Furthermore, DNMT1 has been confirmed as a target of miR-148a and miR-152 [41]. Such a direct effect on the DNA methylation machinery by epigenetically regulated miR NAs could be an indication of an important feedback regulatory mechanism. Several reports have demonstrated that DNMT3a and DNMT3b are fre-quently overexpressed in cancers with poor prognosis, and targets of the miR-29 family, miR-101, miR-143, mir-148a and miR-152 have been reported in various forms of cancer [37,41,43,45,46,47,51,59,61,62,65,66]. Therefore, it appears that all DNMTs are a direct or indirect target of a subset of miRNAs and perform specific functions in the refinement of the expression levels of these crucial epigenetic regulators. As such, dysregulated miRNA expression may contribute to the widespread and often inconsistent changes in DNA methylation patterns detected in cancers, causing both hypo- and hyper-methylation of certain genes and/or regions of the genome [124].

As identified earlier, miR-101 has been reported in a number of cancers, exhibiting direct effects on the epigenetic machinery. Several studies have shown that miR-101 targets EZH2, the catalytic subunit of PRC2, which implies the repressive H3K27me3 signature [38,48,49,59,68]. There is also evidence that miR-128 (downregulated in T-cell leukaemia) [70], miR-138-5p (overexpressed in squamous cell carcinoma) [69], miR-31 (overexpressed in melanoma) [60] also target EZH2. Other important direct epigenetic targets of miRNAs involve HDAC1 and HDAC4 [125,126]. In prostate cancer, HDAC1 is targeted by miR-34b (overexpressed, rendering H3Krme3 modification, also targets HDAC4) [66] and miR-449a (downregulated) [67]. Higher expression of miR-19a, miR-25, miR-32, miR-92b and miR-96 were found to target protein arginine methyltransferase 5 (PRMT5) in leukaemia and lymphoma cells [58]. Nevertheless, it is obvious that an increasing subset of miRNAs is implicated in the regulation of DNA and histone-modifying enzymes (Figure 1), thus highlighting the reputation of these miRNAs in the establishment and maintenance of genomic sustainability and epigenetic architecture.

## 4. In Silico Analysis of Epi-miRNA Associations

To date, only a small subclass of miRNA–mRNA pairs predicted in silico has been experimentally confirmed [127]. The miRNA–mRNA interfaces are based on sequence complementarity (seed match) and have provided opportunities for in silico prediction of target genes for miRNAs of interest [128]. Although there are several diverse factors affecting the ability of miRNAs to recognise and bind their target, when considering general prediction strategies, these features can commonly be categorised into five groups: (i) attributes of the “seed” pairing; (ii) evolutionary conservation; (iii) abundance of the target site; (iv) accessibility of the target sites; and (v) thermodynamic stability of the miRNA–mRNA duplex [127,129,130]. Typically, a combination of these strategies is utilised by most of the currently available prediction algorithms, with some depending heavily on a particular combination, with others serving to balance the prediction mechanism [131]. Future studies in epigenetic regulation of miRNA expression and miRNA mediated epigenetic regulation linked to downstream signalling pathways are likely to lead to the development of novel drug targets in cancer therapy. 

A handful of in silico tools are available and provide information of the regulatory relationship of epi-miRNA with cancer. Functions and link information to the web-application of five selective tools (EpimiR, HMDD, miRCancer, MethyCancer and miRNet) are presented in Table 2. Using these tools, we extracted epi-miRNAs reported in different cancers from EpimiR and HMDD, validated their status using miRCancer and MethyCancer, and finally conducted a network analysis in the miRNet suite (555 miRNAs against 48 epigenetic regulators). The network analysis (Figure 2) identified a list of 101 miRNAs (Table 3; hypergeometric test) to be differentially associated with at least with 1 network (28 miRNAs) and a maximum of 58 networks (miR-515, adj. *p* = 2.55 × 10^11^). 

## 5. Epigenetic Strategies for Cancer Therapy

Epigenetic biomarkers may provide a tool for early disease detection, prognostic indicators and/or cancer prevention through the detection of different or aberrant methylation, histone and expression profiles. Previous studies have identified miRNA signatures of well-established epigenetic miRNAs (epi-miRs) which correlate with overall cancer risk, disease staging and survival [17]. For example, expression signatures of miRNAs have been shown to differentiate between acute lymphoblastic leukaemia (ALL) from acute myeloid leukaemia (AML), subgroups [135,136,137], highlighting their utility as a potential clinical biomarker. Through identification of aberrant hypermethylation of miRNAs, particularly in ALL, promising biomarkers for the prediction of clinical significance have been identified [138]. 

Synthetic oligonucleotides were developed for use as therapeutic agents via in vivo delivery due to their more robust nature against degradation when compared to RNA [139,140]. The five current most common applications used in the regulation of miRNA target gene expression in cancer are reviewed below and include miRNA mimic, anti-miRNA oligonucleotides (anti-miRs), miRNA sponges, miRNA masking and epigenetic inhibitory molecules (Figure 3). These incorporate either the use of oligonucleotide- or virus-based constructs to inhibit oncogenic miRNA or to reactivate a repressed miRNA or tumour suppressor miRNA [22,139,140].

### 5.1. miRNA Mimics and Inhibitors

The use of miRNAs as a potential epigenetic treatment for specific malignancies is an evolving area of research. Several miRNA-based therapeutics (Figure 3) have been studied as potential cancer therapies for both solid tumours and haematological malignancies in specific tumour environments which display different miRNA expression profiles [11,141,142,143]. As epigenetic silencing of miRNAs is involved in the regulation of key pathways such as leukemogenesis, it may provide a target for epigenetic drugs and provide an avenue for inducing re-expression of key regulatory miRNAs. Two main applications used to inhibit tumour development are miRNA mimics and anti-miRs, also known as anti-miRNA oligonucleotides (AMO) [139,140,143,144]. miRNA mimics are made up of synthetic double-stranded RNA which mimic endogenous miRNAs to bind to target gene mRNAs and result in posttranscriptional gene repression of oncogenes as well as to re-express silenced tumour-suppressive miRNAs (tsmiR) or tumour suppressor genes, thereby inhibiting cancer cell proliferation and cell cycle progression [140,145]. As an example, successful application of a miR-218 mimic was used against acute promyelocytic leukaemia cells and showed reduced cell viability and promoted apoptosis [146]. Conversely, tsmiR-497 was found to inhibit breast cancer cell proliferation and disease progression [145]. miRNA mimics have also provided great insight into the functional impact of specific miRNAs in signalling pathways such as JAK/STAT and demethylation, which are commonly implicated in malignant phenotypes [142]. 

Anti-miRs can also alter miRNA-related pathways by binding and blocking oncogenic miRNA access to the mRNA transcript, thereby resulting in either slowed or repressed tumour development, as in the case of anti-miR-126 which resulted in successful inhibition of leukemic cells [147]. miRNA mimics and anti-miRs have the potential to offer personalised miRNA expression therapeutics [141]. The main advantage of epigenetic miRNA (epi-miR) therapeutics would be the ability to regulate multiple pathways through the modification of a single miRNA [5] to provide an adjunct therapeutic agent in the management of specific cancers. Current limitations of this epi-miR technology include the difficulties in ascertaining master regulatory miRNA to prevent unwanted impacts on off-target genes and pathways, along with the low efficiency of these applications due to the variability in tissue and staging-specific miRNA expression [148].

### 5.2. miRNA Sponges

An alternative miRNA inhibitor which can be expressed in cells is termed an miRNA sponge. These inhibitors are transcripts derived from promoter regions which contain multiple common binding sites to the target oncomiR of interest [22]. Vectors encoding these sponges are transiently transfected into cells and allow the sponge to bind to the target oncomiR to inhibit mRNA binding [22]. Sponges are 2–7 nucleotides long and inhibit miRNAs through a complementary heptameric sequence. This allows a single sponge to inhibit an entire miRNA family [22] and has been successfully applied to the repression of oncogenes (Figure 3) in various types of cancers such as leukaemia, sarcoma, breast cancer, renal cancer, lung cancer and melanoma [74,149]. In addition to this, some researchers have successfully used a multi-potent miRNA sponge to simultaneously target and repress multiple oncogenic miRNAs such as miR-155, miR-21 and miR-221/222 in breast and pancreatic cancer cell lines: MDA-MB-436, MCF-7, MIA-Paca-2, Panc-1 and BxPC3 [150].

### 5.3. miRNA Masking

miRNA-masking (miR-mask) applications contain AMO which have been commonly modified with single-chain 2′-O-methyl to increase binding and nuclease activity. miR masks form a 22 nucleotide antisense to an mRNA target of an endogenous miRNA of interest [23], and rather than interacting with the miRNA, they complement and bind to the 3′-UTR site of the target mRNA. This action allows the miR mask to block the endogenous miRNA binding site on the target mRNA disrupting the miRNA inhibitory function. A recent study established that targeting miR-522 led to reduced proliferation of non-small cell lung cancer [151]. Rather than being gene-specific, the effect of miR masks is sequence-specific with potential for adverse side effects and toxicity (Figure 3) [152]. To date, miR-masking efficiency, accuracy, and toxicity still remain inconsistent, making them less suitable for therapeutic use in the treatment of cancers.

### 5.4. Epigenetic Inhibitory Molecules

Epigenetic drug therapies contain inhibitory molecules which target epigenetic machinery such as DNA methyltransferase inhibitors (DNMTi) and histone deacetylase inhibitors (HDACi) [153,154,155]. DNMTi drugs are used to irreversibly inhibit DNMT enzymatic activity and trigger its degradation This application is well established as an epigenetic regulatory agent for the inhibition of epigenetic mutations and oncogene expression [153,154] (Figure 3). A therapeutic application for DNMTis, for example, is to inhibit abnormal tumour gene expression by disrupting key tumour initiation and progression pathways [156]. HDACi alters the acetylation and deacetylation of histone lysines, in which deacetylation is a known contributor to abnormal gene expression in malignancy. Therefore, HDACis can be used to block cell proliferation, promote differentiation and induce apoptosis to reverse cancer initiation and progression [155]. Both DNMTi and HDACi applications have been successfully used in repression of various types of haematological malignancies, however, this application showed variable and limited success in solid tumours [157,158]. 

## 6. Epigenetic Therapeutics in Cancer Clinical Trials

As we have seen by the successful clinical introduction of epigenetic inhibitors like DNMTi and HDACi [153] in the treatment of haematological malignancies [157], epigenetic-based applications are powerful therapeutic agents used in cancer care. To date, several clinical studies have been completed, while others are currently underway investigating the use of miRNA biomarkers in various types of cancer. A phase 1 trial using miR-16 mimic incorporated with epidermal growth factor receptor (EGFR) targeting antibody called TargomiR, was used for patients with either recurrent malignant pleural mesothelioma or non-small cell lung cancer [159,160]. The findings were promising and showed that the miR-16 mimic is beneficial for patients with terminal mesothelioma patients with less than a 10% chance of 5-year survival [161]. A less successful 2013 phase 1 clinical trial investigated the miRNA-34 drug mimic MRX34 in patients with either primary liver cancer, solid tumours, or hematologic malignancy. The study was terminated after some patients experienced serious adverse reactions to the investigational drug [162].

## 7. Overcoming Limitations of miRNA Biomarkers and Therapeutic Agents

A major limitation to the utility of miRNAs as biomarkers for the diagnosis and monitoring of disease progression in malignancy is the natural variability in miRNA expression levels across tissue types and disease stages [163,164,165]. This issue alone makes standardising sample collection methods for use in clinical correlations more challenging. Further to this, once a miRNA biomarker or target gene has been established, a significant limitation to the use of a therapeutic miRNA or epigenetic inhibitory drug is the risk of a serious adverse event due to toxicities like those seen in studies with the miRNA mimic MRX34 and HDAC-based inhibitor drugs [155,162]. As discussed above, this is where the role of rigorous clinical trials has greatly contributed to the recent advancements of miRNA therapeutics. Other significant limitations which are yet to be overcome in some specific miR-malignant cell applications include inefficient delivery, inefficient cellular uptake, short half-life, low intracellular release and low in vivo stability [23,166]. In a clinical setting, non-viral based therapeutic miRNAs may be favoured due to their stable composition, lower immunogenicity and ease of manufacturing [23]. Delivery mechanisms utilising nanoparticles, polymers and liposomes for mediated drug delivery have also been favoured in recent studies [167,168]. However, these applications require further optimisation before miRNAs can become embedded in standard cancer therapeutic development [169]. A possible solution to overcoming these inefficiencies may be direct intra-tumoral injections of miRNA drugs to enhance target efficacy and reduce adverse reactions [144,149,170]. As research continues to advance in this area, several new clinical trials are likely to be conducted to assess the efficiency and reliability of these miRNA therapeutics [11].

In addition, one puzzling phenomenon in cancer treatment is recurrence of cancer cells in a more aggressive manner and avoidance of apoptosis with higher metastatic potential [171]. The reversal of apoptosis is seen in some cancer cells and often results in more aggressive tumours and metastasis. The exact mechanism behind this event is still unknown; however, this reversal process is known to trigger a transition from non-stem cancer cells (NSCCs) to CSCs, which potentially could be suppressed through the use of DNA methylation or demethylation inhibitors before apoptosis induction [171]. A particular state of the tumour cells termed as polyploid giant cancer cells (PGCCs) has been suggested to be responsible for facilitating the escape from therapeutic-induced senescence [172,173]. Tumours can originate from a stem cell via dedifferentiation, therefore, the use of DNMT inhibitors may reactivate tumour suppressor genes, which could disrupt PGCC-mediated dedifferentiation and the development and progression of tumours [174,175]. The level of PGCC increases after exposure to chemotherapeutic drugs like 5-fluorouracil (5-FU) [175]. 5-fluorouracil is used to treat colorectal cancer (CRC), and, interestingly, resistance of CRC to 5-FU has been reported to be associated with the upregulation of nuclear factor-erythroid 2-related factor 2 (Nrf2) via DNA demethylase ten-eleven translocation (TET)-dependent DNA methylation [176]. Therefore, epi-miR based therapeutic strategies should consider the aftermath of any targeted treatment. 

## 8. Conclusions

In this review, we have detailed the relationship between epigenetic alteration of miRNAs and miRNA-mediated epigenetic modifications in cancer. Aberrant DNA methylation generally causes miRNA dysregulation in cancer, and methylation of specific miRNA genes may be a valuable biomarker for cancer diagnosis and prognosis. Variation in the histone architecture also disrupts miRNA expression. Such dysregulation in miRNA expression leads to genome-wide epigenetic abnormalities. It is to be expected that further study of the association between epigenetic regulation of and by miRNAs will lead to the innovative identification and use of new biomarkers as well as therapeutic targets against cancer. The evaluation of the efficiency and reproducibility of miRNA biomarkers across cancer classifications, disease stages and tissues types are also likely to advance in the near future. Reactivation of epigenetic mechanisms of miRNA expression could be an encouraging novel approach of cancer therapy by targeting epigenetically regulated miRNA genes using drugs that inhibit methylation and/or histone modification. Computational analysis of gene regulation is a complex but reliable platform with which to study the epi-miRNA relationship with numerous cellular processes and cancer. These platforms have been successful in identifying epigenetically regulated miRNAs in many malignancies which have paved the way for the use of miRNA-based therapeutics for the treatment of both haematological and solid tumour malignancies. A strategic combination of laboratory-based experimental data, clinical resources and high-throughput computational applications will provide a significant body of knowledge on “epi–miR–epi” regulation for further development of diagnostic and therapeutic suitable for cancer treatments. Further optimisation of delivery applications such as nanoparticle and liposome-mediated delivery, along with additional clinical research trials are required before miRNAs therapeutics can become established as a standard of care therapy in common cancers. With this area of cancer research rapidly evolving, clinical validation of miRNA-based therapeutics may become established following the successful completion of further clinical trials. 

## Figures and Tables

**Figure 1 cancers-12-02922-f001:**
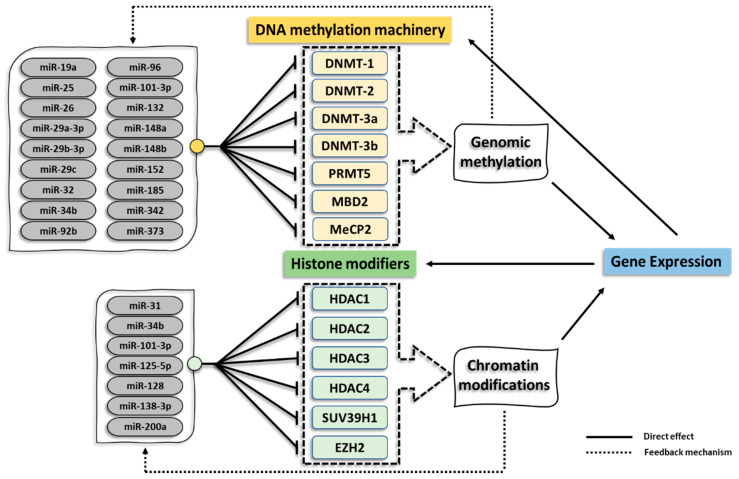
miRNA epigenetic crosstalk. Association of miRNAs with epigenetic regulators involved in the processes of DNA methylation and histone modification. miRNA-mediated alteration of these regulators causes aberrant DNA methylation and chromatin modification. These distorted conditions modify the expression of genes that are involved in modulating the epigenetic machinery and can also affect miRNA expression.

**Figure 2 cancers-12-02922-f002:**
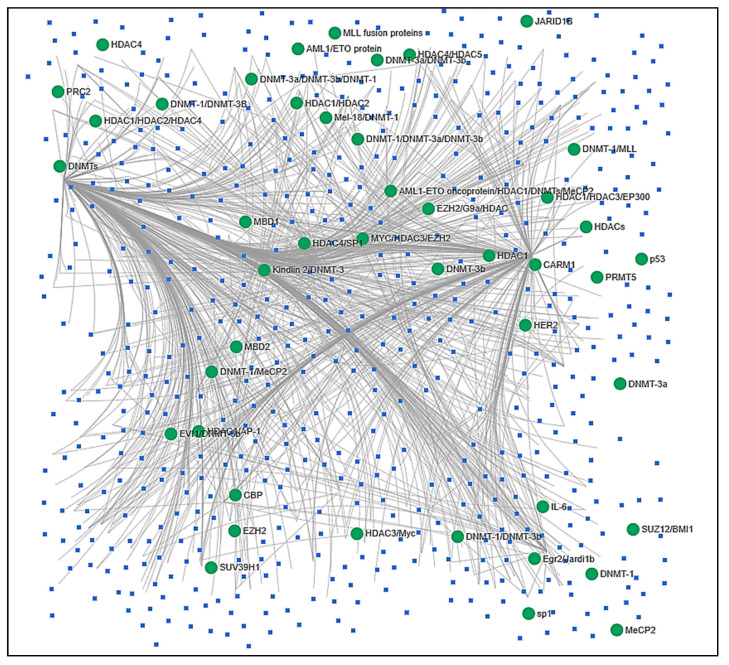
A miRNA-epigenetic network analysis. The diagram represents a network analysis outcome conducted in miRNet suite among 555 miRNAs with 48 epigenetic modifiers hosted by the tool itself. Among those modifiers, DNA methyltransferases (DNMTs) and histone deacetylases (HDACs) showed a strong connection with miRNAs and other modifiers. The enlisted 48 modifiers used were as follows: AML1-ETO oncoprotein/HDAC1/DNMTs/MeCP2, AML1/ETO protein, CARM1, CBP, DNMT-1, DNMT-1/DNMT-3a/DNMT-3b, DNMT-1/DNMT-3b, DNMT-1/DNMT-3B, DNMT-1/MeCP2, DNMT-1/MLL, DNMT-3a, DNMT-3a/DNMT-3b, DNMT-3a/DNMT-3b/DNMT-1, DNMT-3b, DNMTs, Egr2/Jardi1b, EVI1/DNMT-3b, EZH2, EZH2/G9a/HDAC, HDAC1, HDAC1/AP-1, HDAC1/HDAC2, HDAC1/HDAC2/HDAC4, HDAC1/HDAC3/EP300, HDAC3/Myc, HDAC4, HDAC4/HDAC5, HDAC4/SP1, HDACs, HER2, IL-6, JARID1B, Kindlin 2/DNMT-3, MBD1, MBD2, MeCP2, SUZ12/BMI1, Mel-18/DNMT-1, MLL fusion proteins, MYC/HDAC3/EZH2, MYST3, p50 p53, PRC2, PRMT5, RNAPII, sp1, and SUV39H1.

**Figure 3 cancers-12-02922-f003:**
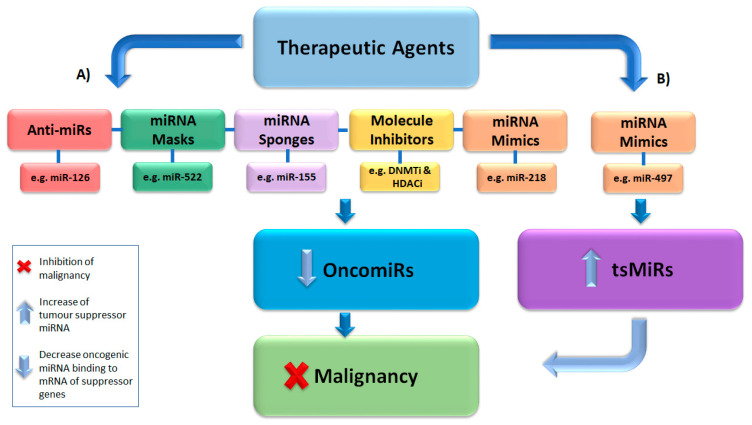
An overview of the therapeutic application of miRNA-based agents used in the prevention of tumour progression. These agents target and regulate the miRNA of interest, resulting in either inhibition of an OncomiR or upregulation of a tumour suppressor gene. Five commonly utilised therapeutic miRNA agents are displayed above showing (**A**) anti-miRs, miRNA sponges and miRNA masks used to suppress oncomiRs by inhibiting miRNA binding to mRNA. miRNA mimics and molecule inhibitor drugs can be used to inhibit DNMT enzymatic activity and trigger its degradation, thereby reducing oncomiR expression and development of malignancy. Similarly, histone deacetylase inhibitors (HDACis) can be used to block tumorigenesis. (**B**) miRNA mimics can also be utilised to mimic the activity of tumour suppressor miRNA in malignancies to reduce or suppress tumorigenesis.

**Table 1 cancers-12-02922-t001:** miRNAs regulate epigenetic modifications in cancer.

Cancer	miRNA	Expression	Epi Regulator	Epi Modification	Epi Target	Ref
Acquired resistance of breast cancer	miR-29a miR-29b-3p	High	DNMT-3a	DNA Methylation	global DNA methylation	[36]
	miR-132	High	MeCP2	DNA Methylation	global DNA methylation	
Acute myeloid leukaemia	miR-29b-3p	High	DNMT-3a/DNMT-3b/DNMT-1	DNA Methylation	ESR1/cyclin-dependent kinase inhibitor 2B	[37]
Bladder transitional cell carcinoma	miR-101-3p	Low	EZH2	H3K27me3	-	[38]
Breast cancer	miR-10a	Low	-	H3K27me3/DNA Methylation	HOXD4	[39]
	miR-29c miR-26b miR-148b	High	DNMT-3b	DNA Methylation	CEACAM6/CST6/SCNN1A	[40]
	miR-148a	High	DNMT-1	DNA Methylation	miR-148a	[41]
	miR-152	High	DNMT-1	DNA Methylation	miR-152	
Breast/Prostate	miR-101-3p	High	EZH2	H3K27me3	miR-101	[42]
Colorectal cancer	miR-143	Low	DNMT-3a	DNA Methylation	-	[43]
	miR-342	High	DNMT-1	DNA Methylation	ADAM23/Hint1/RASSF1A/RECK	[44]
Cutaneous melanoma	miR-29a-3p	High	DNMT-3a/DNMT-3b	DNA Methylation	RASSF1A/TFPI-2/RAR-/SOCS/GATA4	[45]
Endometrial cancer	miR-152	Low	DNMT-1	DNA Methylation	-	[46]
Gastric cancer	miR-148a	High	DNMT-1	DNA Methylation	miR-148a	[47]
Glioblastoma	miR-101-3p	Low	EZH2	H3K27me3	-	[48]
	miR-152	High	DNMT-1	DNA Methylation	miR-152	[49]
Glioma	miR-185	High	DNMT-1	DNA Methylation	ANKDD1A/GAD1/HIST1H3E/PCDHA8/PCDHA13/PHOX2B/SIX3/SST	[50]
Hepatitis B virus (HBV)-related hepatocellular carcinoma	miR-101-3p	High	DNMT-3a	DNA Methylation	RASSF1/PRDM2/GSTP1/RUNX3	[51]
Hepatocellular carcinoma	miR-200a	High	HDAC4	H3ac	miR-200a	[52]
	miR-125b-5p	High	SUV39H1	H3K9me3	Ki67	[53]
	miR-152	High	DNMT-1	DNA Methylation	GSTP1/CDH1	[54]
Hilar cholangiocarcinoma	miR-373	High	MBD2	DNA Methylation	MBD2 regulate RASSF1A	[55]
		High	MBD2	DNA Methylation	miR-373	[56]
Human malignant cholangiocytes	miR-148a miR-152	High	DNMT-1	DNA Methylation	Rassf1a/p16INK4a	[57]
Leukaemia and lymphoma cells	miR-19a miR-25 miR-32 miR-92b miR-96	High	PRMT5	DNA Methylation	H3R8/H4R3	[58]
Lung cancer	miR-101-1	High	EZH2	H3K27me3	CDH1	[59]
Melanoma	miR-31	High	EZH2	DNA Methylation	miR-31	[60]
Multiple myeloma	miR-29b-3p	Low	DNMT-3a/DNMT-3b	DNA Methylation	-	[61]
Non-small-cell lung cancer	miR-29a miR-29b-3p miR-29c	High	DNMT-3a/DNMT-3b	DNA Methylation	FHIT/WWOX	[62]
	miR-29b-3p	High	DNMT-1/DNMT-3a/DNMT-3b	DNA Methylation	PTEN	[63]
	miR-29b-3p	High	DNMT-3b	DNA Methylation	CADM1/RASSF1/FHIT	[64]
Ovarian cancer	miR-185 miR-152	Low	DNMT-1	DNA Methylation	-	[65]
Prostate cancer	miR-34b	High	HDAC1/HDAC2/HDAC4	H3K4me3	miR-34b	[66]
			DNMT-1	DNA Methylation		
	miR-449a	Low	HDAC1	Histone Acetylation	-	[67]
	miR-101-3p	Low	EZH2	H3K27me3	-	[68]
Squamous cell carcinoma	miR-138-5p	High	EZH2	H3K27me3	E-cad	[69]
T-cell leukaemia	miR-101-3p miR-128	Low	EZH2	H3K27me3	-	[70]

**Table 2 cancers-12-02922-t002:** Epi-miRNA databases and networking tools.

Database	Functions	Link	Reference
EpimiR	Contains 1974 regulatory relationships between 19 different epigenetic modifications and 617 miRNAs across *Homo sapiens* and 6 more species. The records are divided into two sections: Epi2miR and miR2Epi.	http://www.jianglab.cn/EpimiR/	[24]
HMDD	The Human microRNA Disease Database (HMDD) collects curated experiment-supported evidence for disease-associated human miRNAs classified into 6 evidence classes (genetics, epigenetics, target, circulation, tissue and other) and 20 evidence codes. It also provides a disease-associated miRNA-target network visualisation function.	https://www.cuilab.cn/hmdd	[132]
miRCancer	The database presently records 878 interactions between 236 miRNAs and 79 human cancers through the processing of more than 26,000 published articles.	http://mircancer.ecu.edu/browse.jsp	[133]
MethyCancer	MethyCancer introduces highly integrated DNA methylation data, cancer-related gene, mutation and cancer evidence from numerous resources, and the CpG island (CGI) clones derived from large-scale sequencing.	http://methycancer.psych.ac.cn/	[134]
miRNet	miRNet enables statistical analysis and functional interpretation of a variety of data produced from existing miRNA studies. The key features include: (i) integration of high-quality miRNA-target interaction data from 11 databases; (ii) differential expression analysis of data from microarray, RNA-seq and quantitative PCR; (iii) flexible options for data filtering, refinement and customisation during network creation; and (iv) a network visualisation system coupled with enrichment analysis.	https://www.mirnet.ca/miRNet/home.xhtml	[25]

**Table 3 cancers-12-02922-t003:** miRNet epi-miRNA network analysis outcome.

miRNA	Hits	*p*-Value	Adj *p*-Value
mir-29	14	0	0
mir-515	58	1.275 × 10^−9^	2.55 × 10^−11^
mir-17	20	1.786667 × 10^−8^	5.36 × 10^−10^
mir-26	9	0.00004275	0.00000171
mir-30	13	0.00024	0.000012
mir-25	10	0.00027	0.0000162
mir-19	8	0.0007728571	0.0000541
mir-193	6	0.0018125	0.000145
mir-290	7	0.006945455	0.000764
mir-9	7	0.006945455	0.000764
mir-34	7	0.006945455	0.000764
mir-196	6	0.02191667	0.00263
mir-664	5	0.02207143	0.00309
mir-129	5	0.02207143	0.00309
mir-10	12	0.02226667	0.00334
mir-15	8	0.03558824	0.0101
mir-101	4	0.03558824	0.0113
mir-1228	3	0.03558824	0.0121
mir-454	3	0.03558824	0.0121
mir-432	3	0.03558824	0.0121
mir-483	3	0.03558824	0.0121
mir-425	3	0.03558824	0.0121
mir-322	3	0.03558824	0.0121
mir-373	3	0.03558824	0.0121
mir-127	3	0.03558824	0.0121
mir-126	3	0.03558824	0.0121
mir-191	3	0.03558824	0.0121
mir-142	3	0.03558824	0.0121
mir-140	3	0.03558824	0.0121
mir-183	3	0.03558824	0.0121
mir-139	3	0.03558824	0.0121
mir-31	3	0.03558824	0.0121
mir-22	3	0.03558824	0.0121
mir-21	3	0.03558824	0.0121
mir-28	5	0.05428571	0.019
mir-365	4	0.06642857	0.0279
mir-146	4	0.06642857	0.0279
mir-221	4	0.06642857	0.0279
mir-218	4	0.06642857	0.0279
mir-132	4	0.06642857	0.0279
mir-27	4	0.06642857	0.0279
mir-24	4	0.06642857	0.0279
mir-124	5	0.08069767	0.0347
mir-663	3	0.09136364	0.0402
let-7	13	0.1133333	0.051
mir-1260a	2	0.1147826	0.0528
mir-135	4	0.1865957	0.0877
mir-130	5	0.2350877	0.119
mir-199	4	0.2350877	0.13
mir-876	2	0.2350877	0.134
mir-542	2	0.2350877	0.134
mir-574	2	0.2350877	0.134
mir-202	2	0.2350877	0.134
mir-324	2	0.2350877	0.134
mir-342	2	0.2350877	0.134
mir-330	2	0.2350877	0.134
mir-361	2	0.2350877	0.134
mir-8	5	0.3482759	0.202
mir-544	2	0.38	0.228
mir-147	2	0.38	0.228
mir-103	4	0.3819672	0.233
mir-7	3	0.4163265	0.272
mir-500	4	0.4163265	0.29
mir-368	4	0.4163265	0.29
mir-194	2	0.4163265	0.325
mir-190	2	0.4163265	0.325
mir-153	2	0.4163265	0.325
mir-219	3	0.4163265	0.343
mir-148	3	0.4163265	0.343
mir-1256	1	0.4163265	0.407
mir-1205	1	0.4163265	0.407
mir-1204	1	0.4163265	0.407
mir-1203	1	0.4163265	0.407
mir-1202	1	0.4163265	0.407
mir-944	1	0.4163265	0.407
mir-943	1	0.4163265	0.407
mir-935	1	0.4163265	0.407
mir-760	1	0.4163265	0.407
mir-765	1	0.4163265	0.407
mir-1323	1	0.4163265	0.407
mir-641	1	0.4163265	0.407
mir-638	1	0.4163265	0.407
mir-636	1	0.4163265	0.407
mir-630	1	0.4163265	0.407
mir-602	1	0.4163265	0.407
mir-601	1	0.4163265	0.407
mir-596	1	0.4163265	0.407
mir-572	1	0.4163265	0.407
mir-568	1	0.4163265	0.407
mir-564	1	0.4163265	0.407
mir-559	1	0.4163265	0.407
mir-498	1	0.4163265	0.407
mir-448	1	0.4163265	0.407
mir-326	1	0.4163265	0.407
mir-375	1	0.4163265	0.407
mir-184	1	0.4163265	0.407
mir-137	1	0.4163265	0.407
mir-302	4	0.4163265	0.408
mir-941	3	0.418	0.414
mir-642	2	0.418	0.418

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
