# Peer review of "Regulatory Mechanisms of Epigenetic miRNA Relationships in Human Cancer and Potential as Therapeutic Targets"

_cancers, 2020, doi:10.3390/cancers12102922_

Round 1

Reviewer 1 Report

Elliott E. K.et.al. Presented by writing this review the Regulatory Mechanisms of Epigenetic-miRNA Relationships in Human Cancer and Potential as Therapeutic Targets. The literature that has been chosen is updated and the review is well-written. Moreover the authors describe in detail how miRNAs regulate all the different epigenetic modifications in cancer and the perspectives to the identification of new drug targets.

Author Response

We thank the Reviewer.

Reviewer 2 Report

The manuscript is interesting, relevant and overall well written. It represents a detailed review on the relationship between between miRNAs and epigenetics in cancer.

Just a comment: bearing in mind that the review article is thought to focus on the role of microRNA, I would avoid the section 5.4. Epigenetic Inhibitory Molecules

Author Response

Authors’ Response:

We thank the Reviewer for their feedback. With regard to section 5.4 Epigenetic Inhibitory Molecules, we believe that in order to offer a comprehensive review as titled “Regulatory Mechanisms of Epigenetic-miRNA Relationships in Human Cancer”, we need to incorporate this section on Epigenetic Inhibitory Molecules and their mechanisms of regulation in various cancers. We believe that removing this section would create a gap in the review surrounding a key aspect in the regulation of epigenetic and miRNA relationships in cancers, as well as resulting in the exclusion of a current and significant target area in cancer therapeutics. To assist in improving the relevance and benefit of section 5.4, we have now added specific examples into lines 394 to 409 on the use of epigenetic inhibitory molecules for the treatment of both Polypoid Giant Cancer Cells and to overcome apoptosis reversal in cancer cells. The new paragraphs include the following content:

“In addition, one puzzling phenomenon in cancer treatment is recurrence of cancer cells in a more aggressive manner and avoidance of apoptosis with higher metastatic potential [172].  The reversal of apoptosis is seen in some cancer cells and often results in more aggressive tumours and metastasis. The exact mechanism behind this event is still unknown however this reversal process is known to trigger a transition from non-stem cancer cells (NSCCs) to CSCs, which potentially could be suppressed through the use of DNA methylation or demethylation inhibitors before apoptosis induction [172].  A particular state of the tumour cells termed as polyploid giant cancer cells (PGCCs) has been suggested to be responsible for facilitating the escape from therapeutics-induced senescence [173,174]. Tumours can originate from a stem cell via dedifferentiation so therefore the use of DNMT inhibitors may reactivate tumour suppressor genes which could disrupt PGCC-mediated dedifferentiation and the development and progression of tumours [175,176]. The level of PGCCs increases after exposure to chemotherapeutic drugs like 5-fluorouracil (5-FU) [176]. 5-fluorouracil is used to treat colorectal cancer (CRC), interestingly, resistance of CRC to 5-FU has been reported to be associated with the upregulation of nuclear factor-erythroid 2-related factor 2 (Nrf2) via DNA demethylase ten-eleven translocation (TET)-dependent DNA methylation [177]. Therefore, epi-miR based therapeutics strategies should consider the aftermath of any targeted treatment.”

Reviewer 3 Report

This is a comprehensive review on the involvement of interplay between miRNAs regulation and epigenetic modulations in various cancers, the application of bioinformatics tools to study these networks, and the promises and challenges of epigenetic-based cancer therapies. I have one major concern (lack of even a mention of the major obstacles in the medical treatment of cancer) and have also noted a few typos.

The authors could summarize the current knowledge on major challenges in cancer therapy and how the subjects covered in this review can help address those challenges. I was hoping to see some discussion on how these networks are linked to any of the responses that contribute to intratumor heterogeneity in terms of therapy resistance. I'm not referring to the conventional (and rather simplistic) thinking of apoptosis and other modes of death, most of which are now known to be reversible, but to responses such as cancer cell dormancy through polyploidy, multinucleation, senescence (PMID: 30040989) and mechanisms that contribute to reversal of death responses (e.g., anastasis for apoptosis) (PMID:29476980).  AS can be seen in these articles (which are just examples of many), epigenetic control plays key roles in these responses (PGCC evolution and apoptotic cancer cell reversal). Thus, a cancer therapy-related manuscript which does not take these key issues into account is incomplete. For example, we known that HDAC inhibitors trigger cancer cell senescence, and this response is reversible and contributes to disease relapse. I'm not suggestion a huge discussion, but at least a mention of these responses and how epigenetic controls impacts them would make this review up-to-date. In short, it is key that the authors at least cite these two key papers and have short statments on how epigenetic control can undelie formation/evolution of Polyploid Giant Cancer Cells (the most deadliest form of cancer cells triggered by anticancer therapy), and reversal of apoptosis. 

I have found only few typos:

Line 61: …role in processes

Line 64: …a hallmark of ??

Line 67: …Delete “it”

Line 89: …(RISC)-induced

Line 102: In the phrase “…therapeutic agents such as those discussed in this review [11,21–23]” perhaps the refs should be placed after “agents”

Author Response

Authors’ Response:

We thank the Reviewer for this constructive feedback and for highlighting a key point to improve the modernity for the manuscript. We have addressed the comments by including new paragraphs outlining the two mentioned phenomena; Polypoid Giant Cancer Cells and apoptosis reversal cancer cells, and how epigenetic controls could be utilised to overcome these events. We have also now added six new references (references 172-177), including the two suggested publications; PMID: 30040989 and PMID:29476980. These new paragraphs have been incorporated into lines 394 to 409 with the below statements:

“In addition, one puzzling phenomenon in cancer treatment is recurrence of cancer cells in a more aggressive manner and avoidance of apoptosis with higher metastatic potential [172].  The reversal of apoptosis is seen in some cancer cells and often results in more aggressive tumours and metastasis. The exact mechanism behind this event is still unknown however this reversal process is known to trigger a transition from non-stem cancer cells (NSCCs) to CSCs, which potentially could be suppressed through the use of DNA methylation or demethylation inhibitors before apoptosis induction [172].  A particular state of the tumour cells termed as polyploid giant cancer cells (PGCCs) has been suggested to be responsible for facilitating the escape from therapeutics-induced senescence [173,174]. Tumours can originate from a stem cell via dedifferentiation so therefore the use of DNMT inhibitors may reactivate tumour suppressor genes which could disrupt PGCC-mediated dedifferentiation and the development and progression of tumours [175,176]. The level of PGCCs increases after exposure to chemotherapeutic drugs like 5-fluorouracil (5-FU) [176]. 5-fluorouracil is used to treat colorectal cancer (CRC), interestingly, resistance of CRC to 5-FU has been reported to be associated with the upregulation of nuclear factor-erythroid 2-related factor 2 (Nrf2) via DNA demethylase ten-eleven translocation (TET)-dependent DNA methylation [177]. Therefore, epi-miR based therapeutics strategies should consider the aftermath of any targeted treatment.”

We have also corrected the typographical errors stated above in lines 61, 64, 67, 89 and moved the in-text references in line 102 to be after the word “agents”.